# A Combination of Transcriptome and Enzyme Activity Analysis Unveils Key Genes and Patterns of Corncob Lignocellulose Degradation by *Auricularia heimuer* under Cultivation Conditions

**DOI:** 10.3390/jof10080545

**Published:** 2024-08-03

**Authors:** Ming Fang, Xu Sun, Fangjie Yao, Lixin Lu, Xiaoxu Ma, Kaisheng Shao, Evans Kaimoyo

**Affiliations:** 1Lab of the Genetic Breeding of Edible Mushroom, College of Horticulture, Jilin Agricultural University, Changchun 130118, China; fangming@jlau.edu.cn (M.F.); lixinl@jlau.edu.cn (L.L.); maxiaoxu54@163.com (X.M.); 2Engineering Research Centre of Chinese Ministry of Education for Edible and Medicinal Fungi, Jilin Agricultural University, Changchun 130118, China; sunxu0512@163.com (X.S.); shaoks217@163.com (K.S.); 3Great East Road Campus, University of Zambia, Lusaka 32379, Zambia

**Keywords:** *Auricularia heimuer*, corncob, transcriptome, CAZymes, lignocellulose

## Abstract

The cultivation of *Auricularia heimuer*, a species of edible mushroom, heavily relies on the availability of wood resources serving as substrate for the growth of the species. To ensure the sustainable development of the *A. heimuer* industry and optimize the utilization of corncob as a substrate, this study sought to investigate the potential use of corncob as a substrate for the cultivation of *A. heimuer*. The purpose of this study was to explore the utilization of corncob lignocellulose by *A. heimuer* at the mycelium, primordium, and fruiting stages, by specifically examining the expression profiles of both carbohydrate-active enzymes (CAZymes) and the transcriptome of differentially expressed genes (DEGs) relevant to corncob biomass degradation. The results revealed 10,979, 10,630, and 11,061 DEGs at the mycelium, primordium, and fruiting stages, respectively, while 639 DGEs were identified as carbohydrate-active enzymes. Of particular interest were 46 differentially expressed CAZymes genes that were associated directly with lignocellulose degradation. Furthermore, the study found that A. heimuer exhibited adaptive changes that enabled it to effectively utilize the cellulose present in the corncob. These changes were observed primarily at the primordium and fruiting stages. Key genes involved in lignocellulose degradation were also identified, including *g6952*, *g8349*, *g12487*, and *g2976* at the mycelium stage, *g5775*, *g2857*, *g3018*, and *g11016* at the primordium stage, and *g10290*, *g2857*, *g12385*, *g7656*, and *g8953* at the fruiting stage. This study found that lytic polysaccharide monooxygenase (LPMO) played a crucial role in the degradation of corncob cellulose, further highlighting the complexity of the molecular mechanisms involved in the degradation of lignocellulose biomass by *A. heimuer*. The study sheds light on the molecular mechanisms underlying the ability of *A. heimuer* to degrade corncob biomass, with implications for the efficient utilization of lignocellulose resources. The findings from this study may facilitate the development of innovative biotechnologies for the transformation of corncob biomass into useful products.

## 1. Introduction

*Auricularia heimuer* is one of the most widely-cultivated edible mushrooms in China, with various beneficial properties, including its content of known anti-oxidants [1], anti-inflammatory constituents [2], and low hypoglycemic index [3]. The species has also been reported to have anti-tumor properties [4] and ability to stimulate immune responses [5]. As a white-rot wood decay, it has the ability to degrade lignin, cellulose, and hemicellulose [6]. This property allows *A. heimuer* to thrive on different agricultural waste substrates, with sawdust being the traditional substrate of choice. The stubborn biopolymer of lignin in wood can be easily modified as the carbon source by *A. heimuer* due to its exceptional capacity to degrade lignin [7]. However, despite its incredible potential and benefits, the cultivation of *A. heimuer* has faced significant challenges. China’s revised forest law has taken effect on 1 July 2020. It has resulted in a decrease in wood supply, causing a surge in wood prices and raising the cost of planting edible fungi.

Cultivation substrate plays a pivotal role in the yield of mushrooms, mycelium growth rate, biological efficiency, and nutritional parameters of edible and medicinal mushrooms [8]. Corn is the most widely planted and highest-yielding crop in the world. Corncob is the major byproduct of the corn industry and is easy to collect from cultivation fields. The ratio of corncob to corn yield is 0.21 [9]. In China, 270 million tons of corn and 55 million tons of corncob were produced in 2021. Most corncobs, as agricultural and sideline products, were burned or stacked for future use. Unfortunately, this leads to air pollution, requirement for storage space, and other environmental burdens in China [10]. The existence of these corncob wastes in the field has brought serious problems to agricultural production. Corncob contains 45% cellulose, 35% hemicellulose, and 15% lignin [11], which can meet the nutritional growth needs of most edible fungi, including *A. heimuer* [12], *Pleurotus ostreatus* [13,14], *Flammulina velutipes* [15], and *Ganoderma lucidum* [16]. Lignocellulose biomass consists of cellulose, hemicellulose, and lignin [17]. The complicated structure of lignocellulose shields it from lignocellulase attack. It forms an obstacle to the transformation of plant biomass. However, filamentous fungi are considered exceptional biomass-degrading agents due to their capacity to secrete all necessary enzymes. There are about 100~1000 carbohydrate-active enzyme genes (CAZymes genes) in different filamentous fungi genomes [18]. These genes code for different products categorized into glycoside hydrolases (GHs), glycosyltransferases (GTs), polysaccharide lyases (PLs), carbohydrate esterase (CEs), carbohydrate-binding modules (CBMs), and auxiliary activities (AAs) [19]. Corncob, which contains a mix of complex carbohydrates, could serve as a substrate and provide adequate nutrients and an appropriate growth environment for edible fungi, which can effectively degrade the lignocellulosic structure of corncobs. As a byproduct of maize, corncob has great potential as a substrate for the cultivation of edible and medicinal mushrooms.

The ability of *A. heimuer* to utilize corncob as a substrate and produce lignocellulose degrading enzymes has been well-established [20]. What remains largely unknown are the specific mechanisms involved in the degradation of corncob biomass. The life cycle of *A. heimuer* is composed of three important stages, namely mycelium, primordium, and fruiting body, in chronological order. They have significant differences in form. To shed light on the process of utilizing corncob, the current research was undertaken to analyze the three stages of transcriptome and extracellular enzyme activity in *A. heimuer* to gain insights into lignocellulose degradation. This may contribute to a significant reduction in the cultivation cost, promote the comprehensive utilization of corncob, and achieve sustainable development of the mushroom cultivation industry. These enzyme activities may reflect both the physiological and biochemical activities of the mycelium and shed light on the degradation products of corncob as a potential cultivation substrate [21,22].

## 2. Materials and Methods

### 2.1. Culture and Sample Collection of A. heimuer

Strain A184 of *A. heimuer* was provided by the Horticultural College of Jilin Agricultural University, Changchun, China. The standard substrate formulation, designated CK, comprised 78% sawdust, 20% wheat bran, 1% lime, and 1% gypsum. The experimental treatment substrate (T1) comprised 78% corncob, 20% wheat bran, 1% lime, and 1% gypsum [23]. Sterile CK and T1 substrates, respectively, were transferred into 90 mm Petri dishes, covered with cellophane, and pure cultures of strain A184 of *A. heimuer* were inoculated onto Petri dishes to obtain sufficient biomass for use in scaled-up cultures in polypropylene bags. Petri dish cultures were kept in the dark at 25 °C, and mycelia were collected from colonies that grew to the brims of the Petri dishes. To sample the primordia and fruiting bodies, each respective substrate with a water content of ~55%–60% was placed into polypropylene cultivation bags (average dry weight 500 g/bag) and sterilized at 121 °C for 2 h. Thirty mushroom bags, each of the treatment (T1) and control (CK), were inoculated with *A. heimuer* strain A184 from the Petri dish cultures in three independent replicates. The *A. heimuer* cultures were left to grow in the dark at 25 °C until the mycelia fully colonized the substrates, after which they were transferred to the cultivation room at 25 °C, 85%~90% relative humidity, scattered light, and good ventilation for fruiting body development. The primordia were collected when the diameter was about 5~10 mm before differentiation of the fruiting body, and the fruiting bodies were collected before spore formation. The collected mycelia, primordia, and fruiting bodies were stored in the refrigerator at −80 °C for RNA extraction. Each biological treatment was conducted in triplicate for RNA-Seq analysis.

### 2.2. Enzyme Extraction and Activity Assay

Five-gram substrates were collected at the stages of mycelium, primordium, and fruiting from treatment 1 and CK in triplicate. To each sample, 45 mL of distilled water was added, and the enzyme liquid was extracted through 8 layers of cotton gauze at 25 °C for 4 h, followed by centrifugation at 4000 r/min for 10 min. The crude enzyme solution was transferred to a sterile centrifuge tube. The activity of β-glucosidase was measured with 2-(hydroxymethyl)phenyl beta-D-glucopyranoside(Phygene Biotechnology, Fuzhou, China). The activity of carboxymethylcellulase (CMCase) was measured with carboxymethylcellulose sodium(Sinopharm Chemical Reagent, Beijing, China). The activity of cellulase was measured with filter paper(Beimu Pulp and Paper, Hangzhou, China). The activity of hemicellulase was measured with xylan(Macklin Reagent, Shanghai, China). The activity of laccase (LACC) was measured with ABTS(Phygene Biotechnology, Fuzhou, China). The activity of lignin peroxidase (Lip) was measured with veratryl alcohol(Macklin Reagent, Shanghai, China). The activity of manganese peroxidase (MnP) was measured with MnSO_4_(Sinopharm Chemical Reagent, Beijing, China). The amount of β-glucosidase, CMCase, and cellulase that generates 1 μ mol of glucose from the substrate per hour was defined as 1 U. The amount of hemicellulase that changed 0.01 OD value was defined as 1 U, under the condition of 1 mL crude enzyme solution reacted with the substrate for 30 min. The amount of amylase that changed the OD value by 0.02 was defined as 1 U, under the condition of 0.2 mL crude enzyme solution reacted with the substrate for 30 min. The amount of manganese peroxidase (MnP), lignin peroxidase (Lip), and laccase (LACC) that catalyzes 1 μmol substrate in 1 min was defined as 1 U. The enzymatic reactions of β-glucosidase, CMCase, cellulase, hemicellulase, and amylase were terminated by adding DNS and then boiling in a water bath. All the enzymes were determined spectrophotometrically [24].

### 2.3. RNA Extraction, cDNA Library Preparation, and Illumina Sequencing

Total RNA from *A. heimuer* samples from the two respective treatments was extracted by the TIANGEN RNAprepPure Plant Kit (Tiangen Biochemical Technology, Shanghai, China) according to the kit manufacturer’s instructions. RNA integrity was assessed using the RNA Nano 6000 Assay Kit of the Bioanalyzer 2100 system (Agilent Technologies, Santa Clara, CA, USA). Enrichment of mRNA with polyA tail by Oligo (dT) magnetic beads, synthesis of cDNA using segmented mRNA as template, random oligonucleotide as the primer, and library construction were all performed according to the normal NEB library building method. After the integrity of the constructed cDNA library was authenticated, samples were sequenced on the Illumina sequencing platform.

### 2.4. Bioinformatics Analysis

To ensure the high quality and reliability of the data for downstream analyses, the original data were filtered through the removal of the reads with adapters, the exclusion of the reads containing indeterminate nucleotides read as N, and the removal of low-quality reads (with base number of Qphred ≤ 20 accounting for more than 50% of the total read length). HISAT2 v2.0.5 software was used to compare the filtered transcriptome clean reads with the reference genome, and StringTie v2.2.0 was used to predict new transcripts. The fast and accurate comparison was conducted by HISAT2 v2.2.1 to connect Clean Reads with the reference genome (https://www.ncbi.nlm.nih.gov/assembly/GCA_002287115.1, accessed on 19 December 2022) and calculate the expression amount for all genes in each sample using FPKM method. The genes with a significant difference in expression level between the two samples were screened by the thresholds of padj ≤ 0.05 and | log_2_FoldChange | ≥ 1. ClusterProfiler v4.0 was used to enrich the screened differential gene sets with KEGG (Kyoto Encyclopedia of Genes and Genomes, https://www.kegg.jp/kegg/pathway.html, accessed on 5 February 2023) and GO (Gene Ontology, https://geneontology.org/, accessed on 5 February 2023) functions. Padj ≤ 0.05 was used as the threshold of significant enrichment for both KEGG and GO pathways. The number of differentially expressed genes (DEGs) in each group was counted, and the enrichment significance of DEGs was detected by the hypergeometric distribution method. The *A. heimuer* transcriptome was analyzed using the CAZy annotation pipeline, and searches against the CAZy database (http://www.cazy.org/Home.html, accessed on 9 February 2023) were performed to functionally annotate carbohydrate-active modules in the annotated genes. Auxiliary activity (AA) modules were identified using dbCAN2 (http://cys.bios.niu.edu/dbCAN2, accessed on 9 February 2023).

### 2.5. RT-qPCR Validation

The transcriptome data were validated by real-time quantitative polymerase chain reaction (RT-qPCR). To this end, total RNA was extracted from mycelium, primordium, and fruiting body samples with Trizol reagent (Phygene Biotechnology, Fuzhou, China). The Transcript All in One First Strand cDNA Synthesis SuperMix for RT-qPCR ((TransGen Biotech, Beijing, China) was used to synthesize the first strand cDNA from the total RNA of different samples of *A. heimuer*. *APRTase* was selected as the reference gene [25]. Primer Premier v5.0 was used to design the primer sequences of the selected differentially expressed genes (Appendix A), and primers were synthesized by the Kumei Biological Company (Changchun, China). The TransStart TOP Green RT-qPCR kit ((TransGen Biotech, Beijing, China) was used for the qRT-PCR reaction, and the relative quantification of gene expression was performed by the 2^−ΔΔCt^ [26]. The preparation of the reaction system and the reaction procedure were all conducted according to the kit manufacturer’s instructions.

## 3. Results

### 3.1. Performance of A. heimuer Strain A184 on CK and T1 Substrates

Visual inspection of the *A. heimuer* strain A184 cultured on T1 and CK substrates showed equivalent vigorous mycelia, primordia, and fruiting body development on both substrates (Appendix A), demonstrating the feasibility of corncob as a viable substrate for *A. heimuer* cultivation under controlled conditions.

### 3.2. Lignocellulose-Degrading Enzyme Activity of A. heimuer

*A. heimuer* strain A184 exhibited comparable trends in extracellular enzyme activity on both T1 and CK, with peak levels observed at similar growth stages. Specifically, the highest activity levels of β-glucosidase, carboxymethyl cellulase, and hemicellulase were detected at the fruiting stage, with the strain exhibiting higher extracellular enzyme activity on T1 than on CK (Figure 1). In contrast, the peak activity levels of cellulase, amylase, and laccase were detected at the primordium stage. Lower cellulase and amylase activities were detected on T1 compared to CK but showed slightly higher activity levels of laccase. The peak activity levels of Lip and MnP were observed at the mycelium stage. Notably, T1 exhibited higher Lip activity levels than CK but lower MnP activity levels.

### 3.3. Quality Evaluation of the Sequenced Data

Clean reads for further analysis were acquired after raw data filtering, checking for sequencing error rate, and GC content distribution (Appendix A). The results showed that the percentage of bases with a Phred value more than 30 (>Q30) in the total bases of each sample was higher than 93%, and high-quality clean reads were obtained for subsequent analysis.

### 3.4. Reproducibility of Results from Triplicate Experiments

Upon completion of the RNA-Seq analysis, correlation analysis was carried out in triplicate biological experiments of CK and T1 to evaluate the repeatability of the experiments. The results from repeat experiments showed that the samples had a close correlation, which indicated the high repeatability of the experiments (Figure 2A). Based on these observations, the RNA-Seq datasets generated in this study were deemed reliable. Additionally, PCA analysis was performed and revealed that the transcript of *A. heimuer* in PCA1 was markedly different between the mycelium, the primordium, and the fruiting stages, respectively (Figure 2B).

### 3.5. RT-qPCR Analysis

Seven DEGs were subjected to RT-qPCR analysis to further validate the transcriptome data of *A. heimuer* grown on the two different substrates. The reaction parameters were stage 1: 95 °C 10 min; stage 2: 95 °C, 15 s, 60 °C 1 min, 40 cycles; melting curve Stage: 95 °C, 15 s, 60 °C, 1 min, 95 °C, 45 s. Remarkably, the RT-qPCR results were in agreement with the gene expression profile obtained from the RNA-Seq data (Figure 3), giving confidence in the validity and reliability of the RNA-Seq data from this study. These findings provide robust evidence that the *A. heimuer* transcriptome data generated in this study are reliable and may be utilized for further analysis.

### 3.6. Differentially-Expressed Genes

At the mycelium, primordium, and fruiting stages, a total of 10,979, 10,630, and 11,061 DEGs were obtained by comparing the transcriptome data of treatment T1 and CK, respectively (Figure 4A). There were 2205, 105, and 856 differentially expressed genes identified at the mycelium, primordium, and fruiting stages, respectively, based on the screening criteria| of log_2_ (FoldChange) | ≥ 1 and padj ≤ 0.05 (Figure 4B). The largest differences in gene expression between T1 and CK were observed at the mycelium stage (JST1 vs. JSTck), followed by the fruiting stage (ZST1 vs. ZSTck), and then the primordium stage (YJ1 vs. YJck).

### 3.7. CAZymes Related to the Decomposition of Corncob Lignocellulose

The transcriptomes of different stages from T1 vs. CK were analyzed to identify the DEGs of CAZymes in *A. heimuer*. A total of 639 DEGs of CAZymes were obtained, of which 92 CAZymes were related to lignocellulose degradation (Appendix A), and 8 CAZymes were related to the degradation of starch (Appendix A). Using the screening criteria of |log_2_ (FoldChange)| ≥ 1 and padj ≤ 0.05, 30 DEGs of the mycelium stage, 6 DEGs of the primordium stage, and 10 DEGs of the fruiting stage were detected (Table 1). One DEG related to starch degradation at the mycelium stage and fruiting stages was identified (Figure 5). There were 14 (down-regulated), 5 (4 up-regulated, 1 down-regulated), and 4 (3 up-regulated, 1 down-regulated) DEGs of CAZymes related to cellulose degradation at the mycelium stage, primordium stage, and fruiting stage, respectively. The ratio of up-regulated genes was 0%, 80%, and 75%, respectively. Additionally, the analysis revealed 6, 1, and 0 DEGs of CAZymes related to hemicellulose degradation at the mycelium stage, primordium stage, and fruiting stage, respectively. The six DEGs at the mycelium stage included one xylanase (XYN, down-regulated), three arabinofuranosidase (ABFs, two down-regulated, one up-regulated), and two acetyl xylan esterase (AXEs, down-regulated) at the mycelium stage (Table 1A, Appendix A). Moreover, 9, 0, and 5 DEGs of CAZymes related to lignin degradation were identified at the mycelium stage, primordium stage, and fruiting stage, respectively. The nine DEGs at the mycelium stage included four laccases (LACCs)—three down-regulated and one up-regulated; one down-regulated versatile peroxidase (VP); one up-regulated pyranose oxidase (PYO); one down-regulated alcohol oxidase (AOX); and two glyoxal oxidase (GLOX)—one down-regulated, the other up-regulated—at the mycelium stage (Table 1A). The five DEGs at the fruiting stage included one up-regulated LACC, one down-regulated PYO, one up-regulated AOX, and two GLOXs, both down-regulated (Table 1C). These findings provide important insights into the molecular mechanisms of CAZyme-mediated lignocellulose degradation in *A. heimuer*.

According to the results from *A. heimuer* transcriptomes on T1 vs. CK, 13 up-regulated DEGs were screened out, including 4 genes (*g6952*, *g8349*, *g12487,* and *g2976*) at the mycelium stage, 4 genes (*g5775*, *g2857*, *g3018,* and *g11016*) at the primordium stage, and 5 genes (*g10290*, *g2857*, *g12385*, *g7656,* and *g8953*) at the fruiting stage. Noteworthy among these were *g12487,* encoding PYO at the mycelium stage, and *g2857* and *g10290,* both encoding LPMO at the primordium and fruiting stage. Their expression levels were 42.2, 13.2, and 4 times higher on T1 than CK (Appendix A). These DEGs were the key genes in the degradation process of lignocellulose in the corncob at different growth stages.

### 3.8. KEGG Pathway Enrichment Analysis

Using KEGG tools to scrutinize the decomposition of corncob and sawdust lignocellulose at three distinct growth stages, a total of 93, 10, and 66 pathways were enriched at the mycelium, primordium, and fruiting stages, respectively. Nevertheless, the pathway related to lignocellulose degradation was the carbohydrate metabolism pathway, which consisted of three metabolic pathways, namely glycolysis/gluconeogenesis (adl00010), pentose and glucuronate interconversions (adl00040), and starch and sucrose metabolism (adl00500).

Compared with the CK, the glycolysis/gluconeogenesis pathway of T1 involves *g2222* encoding hexokinase (EC:2.7.1.1), which was up-regulated at the mycelium stage (Appendix A) and *g11529* encoding aldehyde dehydrogenase (NAD+) (EC:1.2.1.3), which was up-regulated at the fruiting stage (Appendix A). In the mutual transformation pathway of pentose and glucuronic acid, the expression of g3872 encoding L-iditol 2-dehydrogenase (EC:1.1.1.14) was up-regulated at the fruiting stage (Appendix A). The DEGs up-regulated by the above two pathways were involved in carbohydrate metabolism, which was required for the growth and development of the *A. heimuer* fruiting body.

In starch and sucrose metabolism (Figure 5), *g3952* encoding 1,3-β-glucan synthase (EC:2.4.1.34) and *g2222* encoding hexokinase (EC:2.7.1.1) were up-regulated at the mycelium stage (Figure 5A), *g6446* encoding α,α-trehalase (EC:3.2.1.28) was up-regulated at both mycelium and fruiting stages (Figure 5B), *g7476* and *g11536* encoding 1,3-β-glucosidase (EC:3.2.1.58) was up-regulated at the mycelium stage, and *g2336*, *g7476,* and *g6956* genes encoding this enzyme were up-regulated at the fruiting stage. At the mycelium stage, *g9825,* encoding glycogen phosphorylase (EC:2.4.1.1), was up-regulated. However, these genes do not belong to the core gene of CAZymes. Four CAZymes candidate genes were involved in this pathway. At the mycelium stage, *g6265*, *g564*, and *g10166* encoding CBHs (EC:3.2.1.91) were down-regulated. At the primordium stage, *g5775* encoding EGL (EC:3.2.1.4) was up-regulated, consistent with the above DEG analysis (Appendix A).

### 3.9. GO Annotation

At the mycelium stage, the difference was mainly concentrated in Molecular Function (MF). Carbohydrate binding (GO: 0030246) enriched 28 DEGs, of which *g5776* and *g5878* were up-regulated, and the rest of the 26 genes were down-regulated. In cellulose binding (GO: 0030248), 17 DEGs were enriched. However, all of them in this function were down-regulated. Forty-eight DEGs were enriched in redox activity (GO: 0016705), of which 21 up-regulated genes were involved in cofactor binding. In addition to *g3728* and novel 860, 19 other genes, such as *g9514* and *g3447*, all participate in iron binding and heme binding. Among them, Fe (II) reacted with hydrogen peroxide to form hydroxyl radical, which was a highly active oxidant capable of depolymerizing cellulose. At the fruiting stage, carbohydrate metabolism (GO: 0005975) enriched 36 DEGs, of which 9 genes were up-regulated. The activity of the hydrolase (GO: 0004553) (GO: 0016798) acting on the glycosidic bond in MF was enriched with 28 DEGs, of which 8 genes were up-regulated. These eight genes were classified into seven glycosidic hydrolase families, namely GH2, GH5, GH16, GH25, GH37, GH43, and GH81.

## 4. Discussion

### 4.1. The Extracellular Enzyme and the DEGs of CAZymes

The chemical makeup and molecular structure of corncob lignocellulose have been reported to differ significantly from that of sawdust, with distinct composition ratios. Specifically, corncob lignocellulose comprises 45% cellulose, 35% hemicellulose, and 15% lignin [11], while hardwood contains 38%~51% cellulose, 17%~38% hemicellulose, and 21%~31% lignin [27]. The overall changes in extracellular enzyme activities across various substrate cultivations exhibited a parallel trend (Figure 1). It indicated that the effects of different substrates on the degradation of lignocellulose were mainly on the expression levels of related genes rather than a significant impact on the pattern.

In general, the lignin in the plant cell wall was degraded first by oxidation or modification, which exposes cellulose and hemicellulose for further use by fungi [28]. A pairwise comparison of DEGs between the three growth stages of T1 showed that lignin peroxidase (LiP) and manganese peroxidase (MnP) activities were highest at the mycelium stage and gradually decreased at the primordium and fruiting stages, consistent with the structure of lignocellulose. According to the results of transcriptome analyses, 10 of 13 DEGs coding ligninase were down-regulated in the transcriptome of YJT1 vs. JST1, and 9 of 13 DEGs were down-regulated in the transcriptome of ZST1 vs. JST1 (Appendix A). The ratio of down-regulated DEGs coding ligninase was more than 70% at the primordium and fruiting stages. This demonstrated that the trend of changes from transcriptome was consistent with that of extracellular enzyme activities. Most of the DEGs related to the degradation of cellulose and hemicellulose were up-regulated at the mycelium stage, compared with the primordium stage and fruiting stage (Appendix A). This result was different from the changing pattern of the extracellular enzyme activities. The hydrolysis of cellulose and hemicellulose requires the synergistic action of many types of enzymes [29]. The limited types of extracellular enzymes analyzed in this study cannot fully reflect the overall changes in cellulase and hemicellulase activities. Therefore, it is recommended to explore more types of related enzymes in future research, such as pectinases, chitinases, esterases, and lytic polysaccharide monooxygenases (LPMO), for more accurate results. Corncob has a higher content of cellulose and hemicellulose than sawdust [11,27], so genes related to substrate degradation showed higher expression levels at the mycelium stage. Based on the above results, lignocellulose-degradation enzymes were most active at the mycelium stage cultivated on corncob.

A pairwise comparison of DEGs between corncob and sawdust showed the enzymatic hydrolysis rate of corncob cellulose was lower at the mycelium stage. This could be attributed to the fact that the crystallinity index of cellulose is directly related to the initial enzymatic hydrolysis rate of cellulose, as in Sun’s research [30]. A comparison of DEGs to that from CK showed 7 of the 9 DEGs of CAZyme at the primordium and fruiting stages to be up-regulated. The seven up-regulated DEGs included five LPMOs, accounting for 71.4% of all up-regulated genes. It is worth mentioning that the fold-change of up-regulated DEGs was higher than that of down-regulated DEGs, particularly *g2857*, whose fold-change was 2^3.7^, with the greatest changes appearing at the primordium stage (Appendix A). The important role of LPMOs in the decomposition of corncob cellulose was highlighted, as it was found to be capable of initiating the cracking of crystalline cellulose and thus improving the activity of lignocellulolytic enzymes [31,32]. This result was similar to the transcriptome expression pattern of *Agaricus bisporus* (straw-rotting fungus) cultivated on straw [33]. In addition, this result was generally consistent with the results of the extracellular enzyme activity, indicating that *A. heimuer* showed an adaptive ability to utilize the cellulose in the corncob, even though the change appeared at the primordium stage and the fruiting stage, presumably caused by cellulose induction. Among the hemicellulose-degrading enzymes, only one DEG was up-regulated at the mycelium stage, while five DEGs were down-regulated, and one DEG was down-regulated at the primordium stage. No DEG was found at the fruiting stage. This result was similar to the variated pattern of extracellular enzyme activity at the primordium stage and fruiting stage but was opposite to the extracellular enzyme activity of hemicellulose at the mycelium stage. The degradation of hemicellulose is very complex and requires a set of hemicellulases. The only type of hemicellulase determined in this research might account for the differences in results, as many enzymes are involved in hemicellulose degradation. In addition, the substitution of sawdust by corncobs did not affect the transcription level of hemicellulase at the yield-forming stage, as there was no significant difference in the expression levels of T1 and CK at the fruiting stage. Regarding lignin-degrading enzymes, 14 DEGs were screened out at the mycelium and fruiting stages, with nine DEGs identified at the mycelium stage, of which seven were down-regulated and two were up-regulated. Among the screened DEGs, *g12487* encoding PYO had the highest expression level in T1, which was 41.4 times that of sawdust. At the fruiting stage, five DEGs were screened out, of which three were down-regulated, and two were up-regulated. The results of DEGs were generally consistent with those of extracellular enzymes, confirming the important role of lignin-degrading enzymes in the decomposition of corncob cellulose.

Compared with the results of extracellular enzymes and the DEGs, some of the results were not consistent. For instance, the DEGs of T1, which are responsible for cellulose and hemicellulose degradation, showed down-regulation at the primordium and fruiting stages. However, β-glucosidase, carboxymethyl cellulase, and hemicellulase exhibited peak activity at the fruiting stage. Thus, to explore the variation tendency of lignocellulose degradation, future research should use more extracellular enzymes to determine their activities.

### 4.2. The Pattern of Lignocellulose Degradation

In this study, 639 CAZymes associated with biomass catabolism were obtained in both T1 and CK transcripts. These results are similar to those on transcriptome analysis of *Pleurotus djamor* that was cultivated on corn stalk, where a total of 675 CAZymes were annotated [34]. On the other hand, in the transcriptome of *Lentinus edodes* that was cultured on substrates containing glucose and cellulose, 461 CAZymes were annotated [35]. The reasons for the above results, besides the differences between substrate and fungal species [36], may partly be explained by the fact that some of the CAZymes genes were induced by specific components in different substrates. According to the results of this study, more cellulose-coding genes were expressed than hemicellulases and ligninases at all three growth stages, indicating that cellulose degradation was most active in *A. heimuer* under corncob cultivation.

Between the pairwise comparisons of the T1 growth stages, the dominant CAZyme class was all GHs. They were also the largest group among 509 CAZymes enriched in the whole genome of *A. auricula,* as reported by [37]. When delving into the T1 expression level of EGL at different growth stages associated with cellulose degradation, it was found that the order of expression levels was JST > ZST, > YJ, and as for the expression level of CBH, it was JST > YJ > ZST. The expression level of BGL was ZST > JST > YJ [38]. It can be deduced that cellulose degradation of T1 was more active at the mycelium stage than at the other stages.

Some genes that were not previously identified as CAZymes were found in this study, and these may play a role in the degradation of lignin. For instance, the expression level of *g865* in samples on T1 was nearly 256 times higher than that of CK at the mycelium stage. This gene encodes aromatic peroxidase (APO), an extracellular enzyme that catalyzes the oxyfunctionalization reactions of aromatic and aliphatic hydrocarbons. Moreover, it was independent of a cellular environment. Due to their high degree of glycosylation, they are quite stable and soluble in aqueous environments [39]. The first APO was discovered in the widely cultivated agaric basidiomycete *Agrocybe aegerita* [40]. Additionally, a Dyp-type peroxidase gene *g1419* was also discovered in T1, and its expression level was seven times higher than that of CK. This gene is relatively new compared to the AA1 and AA2 families, and it has various functions, including oxidative and hydrolytic activity [41,42]. The presence of an alternative biocatalytic system for the direct oxidation of recalcitrant methoxylated aromatics within the lignin polymer was also reported by Liers, which could complement or replace the “classic” high-redox potential peroxidases (LiP, VP) [43]. These genes are worthy of attention and further investigation in future research.

The correlation between the 13 up-regulated DEGs of different growth stages and the performance of agronomic traits is worthy of further study. The 13 DEGs have the potential as the key molecular makers for the breeding of new corn varieties that are suitable for the cultivation of *A. heimuer* by corncob. Especially DEG *g12487* at the mycelium stage and *g2857* at the primordium stage and fruiting stage. Besides the up-regulated DGEs, some down-regulated DEGs also should be further considered. These include *g6265*, *g564*, *g10290*, *g564*, *g2855*, *g2857*, *g7717*, *g12385* and *g3295* at the mycelium stage, *g19* at the primordium stage, and *g6465* at the fruiting stage. These genes were also putatively related to the adaptive cultivation of *A. heimuer* on corncob. Additionally, the genes coding LPMO that were down-regulated at the mycelium stage but up-regulated at the primordium stage and fruiting stage dramatically could also be analyzed to understand the molecular basis of such differential gene expression. The LPMO gene product is related to the degradation of cellulose, and the expression level of different LPMO coding genes changed considerably at different stages. It would be informative to additionally explore in future studies the presumed adaptive changes and determine whether the expression level of such genes could reflect the ability of corncob degradation.

### 4.3. The KEGG Pathway and the DEGs of CAZymes

According to the padj ≤ 0.05, DEGs were enriched in the biosynthesis of secondary metabolites and steroid biosynthesis at the mycelium stage. In the fruiting stage, DEGs were enriched in valine, leucine, and isoleucine degradation. Research has shown that reducing isoleucine intake may improve the expression of genes related to glycolysis, causing obvious metabolic benefits. Isoleucine may be the essential amino acid, allowing a low-protein diet to improve metabolism [44]. Among the six up-regulated genes from T1 vs. CK comparisons, g12710 exhibited a 3.36-fold change and may have a positive effect on the corncob’s metabolism. In the starch and sucrose metabolic pathway, g9825 encoding glycogen phosphorylase was significantly up-regulated at the mycelium stage and may be involved in transforming starch into D-glucose 1-phosphate. The activity of amylase was higher in T1 than in CK at the mycelium stage, indicating that g9825 might be a critical gene responsible for amylase degradation in corncob. Additionally, in the results of KEGG annotation, 138 novel genes were discovered; 107 genes were involved in the metabolism pathway, 25 genes were involved in genetic information processing, and 6 genes were involved in cellular processes. Two genes were involved in both the glycolysis/gluconeogenesis pathways and galactose metabolism pathways (gene ID:novel.828 and novel.1026), one was involved in the interconversion of pentose sugar and glucuronic acid (gene ID: novel.883), one was involved in starch and glucose metabolism (gene ID: novel.1391), and one was involved in the degradation of glycans (gene ID: novel.1451). These novel genes may be promising candidates for further research on related metabolic pathways.

RNA-seq, as a gene sequencing technology, has become the primary method of transcriptome research due to its high throughput, high sensitivity, and response range. In this study, RNA-seq was used to analyze the transcriptome of *A. heimuer* cultivated on two different substrates. The findings have yielded a comprehensive dataset of genes encoding CAZymes in corncob, providing a solid foundation for elucidating the degradation mechanism of *A. heimuer* in corncob cultivation. These results could pave the way for a part replacement of sawdust and facilitate the biotransformation of corncob biomass resources.

## Figures and Tables

**Figure 1 jof-10-00545-f001:**
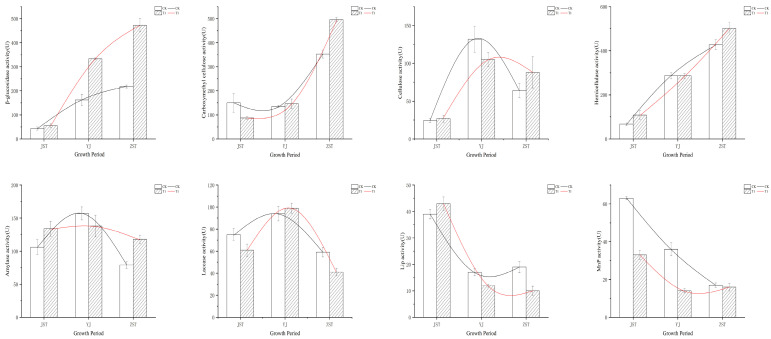
Extracellular enzyme activities of *A. heimuer* at three growth stages. JST: mycelium stage; YJ: primordium stage; ZST: fruiting stage. The black line and white bar represent CK, the red line and grid bar represent treatment 1.

**Figure 2 jof-10-00545-f002:**
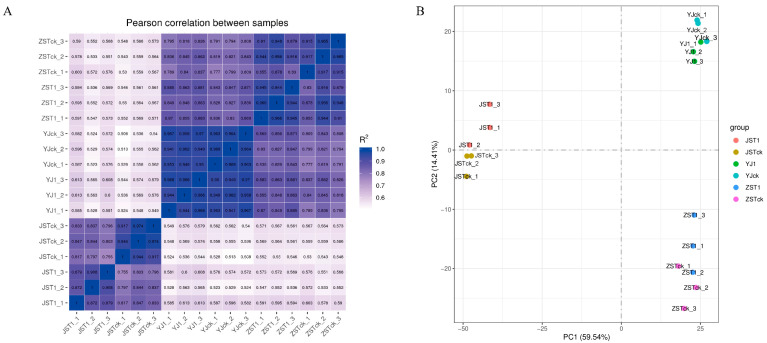
Sample relationship analysis. (**A**): Correlation analysis heat diagram of three biological experiments for each sample of CK and T1 groups. (**B**): The top two principal components (X-axis, PC1; Y-axis, PC2) deciphered 83.95% of all. Note: ZST1, YJ1, and JST1 represent the samples of fruiting body, primordium, and mycelium. T represents the treatment of the test, ck represents CK, and the numbers represent the replicates from 1 to 3 accordingly.

**Figure 3 jof-10-00545-f003:**
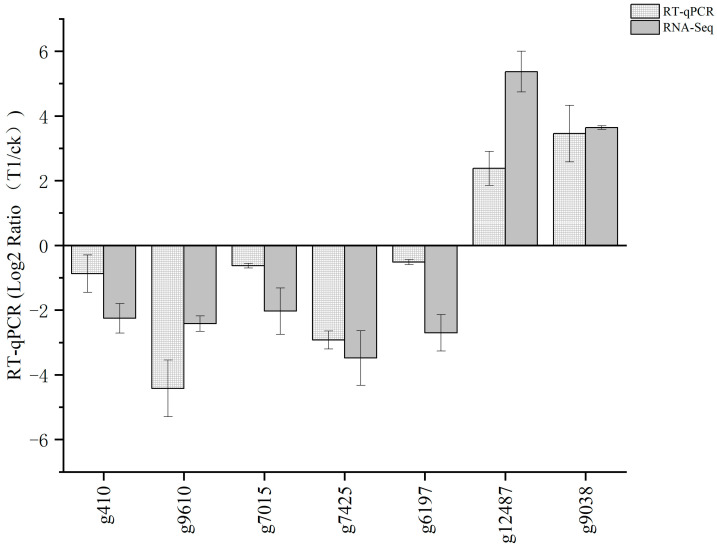
Validation of the RNA-Seq of 7 selected DEGs in the *A. heimuer* transcriptome by RT-qPCR. APRTase was chosen as the reference gene. The Light grey column represented the log2 Ratio (T1/CK) of DEGs, as calculated by the RT-qPCR analysis. The dark grey column represented the log2 Ratio (T1/CK) of DEGs, as calculated by the RNA-Seq analysis. The error bars represented the standard deviation of three biological replicates and three replicates of RT-qPCR runs.

**Figure 4 jof-10-00545-f004:**
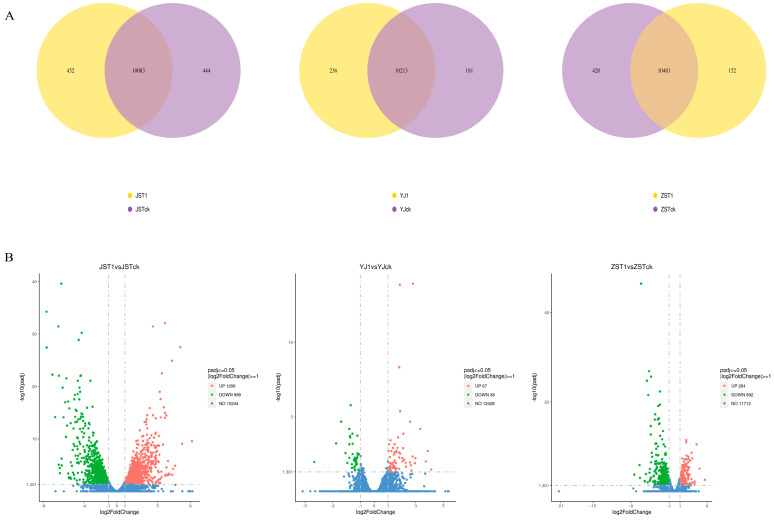
DEGs at three stages. (**A**): Yellow represents the T1-specific gene, purple represents the CK-specific gene, and the overlapping area is the common gene between T1 and CK. (**B**): DEG volcano map. Red represents the significantly up-regulated genes, green represents the significantly down-regulated genes, and blue represents no significant difference in genes.

**Figure 5 jof-10-00545-f005:**
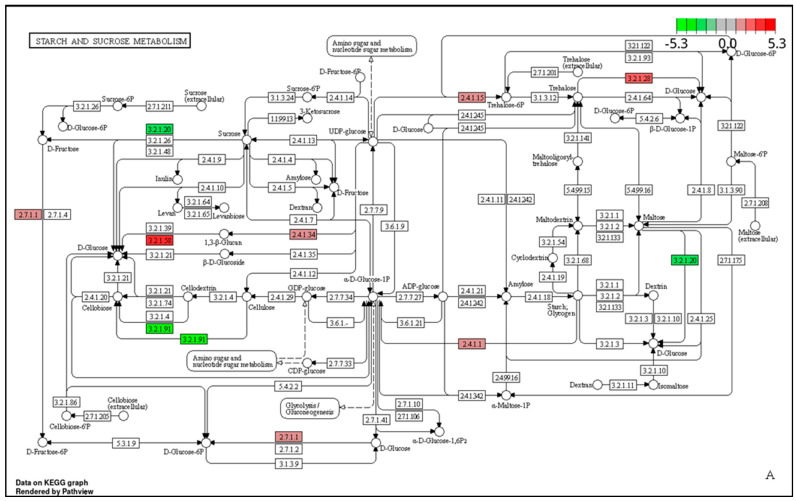
The starch and sucrose metabolism pathway (adl00500) of T1. The reaction in the starch and sucrose metabolism. (**A**): The mycelium stage. (**B**): The fruiting stage. The number in the frame represents the enzyme number, which was stipulated by the Enzyme Commission. The red and green frames indicate that the enriched genes were up-regulated or down-regulated. The circle represents the metabolite, and the arrow represents the enzymatic reaction.

**Table 1 jof-10-00545-t001:** The putative CAZy DEGs from the transcriptome of T1 vs. CK. (**A**). Lignocellulose and starch degradation at the mycelium stage. (**B**). Lignocellulose degradation at the primordium stage. (**C**). Lignocellulose and starch degradation at the fruiting stage.

log_2_(FoldChange)	Gene ID	EC NO.	CAZy Family	Enzyme Name	Substrate
(**A**)					
−3.41	g9204	3.2.1.4	GH5	EGL	Cellulose
−2.82	g10166	3.2.1.4	GH7	EGL	Cellulose
−5.28	g6265	3.2.1.91	GH6	CBH	Cellulose
−4.79	g564	3.2.1.91	GH6	CBH	Cellulose
−6.07	g10290	1.14.99.54	AA9	LPMO	Cellulose
−3.97	g2855	1.14.99.54	AA9	LPMO	Cellulose
−4.82	g2857	1.14.99.54	AA9	LPMO	Cellulose
−2.09	g7717	1.14.99.54	AA9	LPMO	Cellulose
−5.22	g12385	1.14.99.54	AA9	LPMO	Cellulose
−4.86	g3295	1.14.99.54	AA9	LPMO	Cellulose
−2.07	g11918	3.2.1.21	GH1	BGL	Cellulose
−1.86	g11016	3.2.1.21	GH3	BGL	Cellulose
−1.15	g6414	3.2.1.21	GH3	BGL	Cellulose
−2.96	g7098	3.2.1.21	GH3	BGL	Cellulose
−5.64	g19	3.2.1.8	GH11	XYN	Hemicellulose
1.70	g6952	3.2.1.55	GH51	ABF	Hemicellulose
−2.10	g6868	3.2.1.55	GH51	ABF	Hemicellulose
−2.36	g1086	3.2.1.55	GH62	ABF	Hemicellulose
−4.39	g6465	3.1.1.72	CE1	AXE	Hemicellulose
−3.36	g9386	3.1.1.72	CE16	AXE	Hemicellulose
−2.10	g2170	1.10.3.2	AA1	LACC	Lignin
−1.50	g6647	1.10.3.2	AA1	LACC	Lignin
2.32	g8349	1.10.3.2	AA1	LACC	Lignin
−2.82	g7656	1.10.3.2	AA1	LACC	Lignin
−1.85	g8920	1.11.1.13	AA2	MnP	Lignin
5.37	g12487	1.1.3.10	AA3_4	PYO	Lignin
−2.16	g9621	1.1.3.13	AA3_3	AOX	Lignin
1.13	g2976	1.2.3.15	AA5_1	GLOX	Lignin
−1.22	g4342	1.2.3.15	AA5_1	GLOX	Lignin
1.76	g9825	3.2.1.33	GH13	AMY	Starch
(**B**)					
1.19	g5775	3.2.1.4	GH5	EGL	Cellulose
3.72	g2857	1.14.99.54	AA9	LPMO	Cellulose
1.62	g3018	1.14.99.54	AA9	LPMO	Cellulose
−1.97	g12385	1.14.99.54	AA9	LPMO	Cellulose
1.25	g11016	3.2.1.21	GH3	BGL	Cellulose
−1.40	g6868	3.2.1.55	GH51	ABF	Hemicellulose
(**C**)					
2.15	g10290	1.14.99.54	AA9	LPMO	Cellulose
1.44	g2857	1.14.99.54	AA9	LPMO	Cellulose
1.98	g12385	1.14.99.54	AA9	LPMO	Cellulose
−1.27	g2977	1.14.99.54	AA9	LPMO	Cellulose
1.95	g7656	1.10.3.2	AA1	LACC	Lignin
−1.93	g12487	1.1.3.10	AA3_4	PYO	Lignin
1.93	g8953	1.1.3.13	AA3_3	AOX	Lignin
−2.28	g2976	1.2.3.15	AA5_1	GLOX	Lignin
−1.22	g4342	1.2.3.15	AA5_1	GLOX	Lignin
−2.47	g11660	3.2.1.1	GH13	α-amylase	Starch

Note: The strip color from blue to red represents the extent of significantly down-regulated and up-regulated expression in the T1 *A. heimuer* transcriptome compared to that in the CK transcriptome. The numbers represented the ratio of log2(T1/CK).

## Data Availability

The data presented in this study are openly available in NCBI at https://www.ncbi.nlm.nih.gov/bioproject/942987, reference number PRJNA942987.

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
