# Peer review of "A Combination of Transcriptome and Enzyme Activity Analysis Unveils Key Genes and Patterns of Corncob Lignocellulose Degradation by Auricularia heimuer under Cultivation Conditions"

_jof, 2024, doi:10.3390/jof10080545_

Round 1

Reviewer 1 Report

Very nice and interesting study! 

Can there be more information added to the intro about the primordium, fruiting stage, and mycelium stages and the differences associated with each?

If it’s a standard formulation of media, then it should have a citation. I personally am unfamiliar with CK media.

Please cite all protocols for the enzyme assays for b-glucosidase, CMCase, cellulase, hemicellulose, amylase, MnP, LiP, and laccase.

How were the designed primers verified?

Intro: English needs to be edited.

“As a white wood rot fungus” should be corrected to white-rot wood decay fungus

“this unique property” it’s not unique all white-rot fungi can degrade lignin, cellulose, and hemicellulose

“Lignin, the stubborn biopolymer” please do not anthropomorphize a compound

“cultivation of A. heimuer’s” A. heimuer should not be possessive

The coming into effect of the China Forest Law? English needs to be addressed

Materials and Methods:

Was the corn cob macerated for the growth media?

Why was APRTase used as the housekeeping gene?

Please cite Livak and Schmittgen for using their analysis method for RT-qPCR

What were the parameters of the RT-qPCR protocol?

Results:

It’s very hard to read Figure 1. Can the legend be moved to the text? I can not see what the red and black lines signify. I know it’s supposed to be CK and T1 but again it’s very hard to see.

Figure 2: does ZST1 correspond to fruiting body, YJ1 primordium, and JST1 mycelium? The figure text is a bit confusing.

3.5 heading can be RT-qPCR instead of RT-qRT-PCR

Discussion: English edits are necessary

“the lignin in the plant cell wall” should read lignin in the wood cell wall if you are talking about sawdust (which this is unclear)

Can you expand more on why LiP and MnP activities were highest at the mycelium stage? Is this consistent across other white-rot fungi on sawdust?

Why was the ratio of downregulated ligninase DEGs more than 70% at the primordium and fruiting body stages? Please explain further.

“hydrolysis of cellulase… many types of enzymes” such as???

Some citations didn’t convert to numerical representation in discussion.

“mycelium stage cult” ? what does that mean

Any thoughts as to why GH were the dominant CAZys on T1?

“cellulose degradation of T1 was more active at the mycelium stage” Please explain further. Why would the mycelium cause higher cellulose degradation than primordium or fruiting?

Do other studies have information on APO that can be discussed?

Very interesting about g1419!

KEGG annotation of 138 novel genes is also very interesting! It would be nice if there was a bit more discussion on this.

Author Response

Dear reviewer, thank you very much for your kind help and patience, all the questions have been numbered and revised accordingly. Please check the details in my revised manuscript and the attached file.

Reviewer 2 Report

In this work, the decomposition of lignocellulose by fungi was studied. The data obtained are of scientific interest and shed light on the participation of enzymes in the destruction of corncob. The data is presented clearly, except for a few points (see notes). You need to check the formatting of the article. There are different fonts, line spacing, etc.

Unfortunately the lines are not numbered. Therefore, it is difficult to navigate through the text.

page 2. “Lignocellulose biomass consists of cellulose, hemicellulose, and lignin [17] with the latter composed of a solid and complex matrix of cellulose and hemicellulose. The mix of complex modified polysaccharides shields the cellulose and hemicellulose from lignocellulase attack and is an obstacle to the transformation of plant biomass” - Unclear. Does lignin consist of cellulose and hemicellulose? What is the “mix of complex modified polysaccharides” that shields cellulose and hemicellulose? What is a “lignocellulase” (singular)? These points should be clarified.

page 3. Section “Enzyme extraction and activity assay” This section should be rewritten in more detail. It would be necessary to describe by what specific method and with what substrate the enzymes activity was measured. What was taken as the unit of enzyme activity? Reference 23 is given for a detailed description of the methods. The article is in Chinese. Unfortunately the English version was not found. “Enzyme-catalyzed reactions were terminated by 3,5-dinitrosalicylic acid (DNS)…” All enzymatic reactions were terminated by DNS?

page 5. Figure 1. The color of the picture is quite dull, it could have been clearer. The y-axis shows activity in units. Is this specific activity? This should have been indicated.

p 12 “Corncob has a higher content of cellulose and hemicellulose than sawdust (Sun and Cheng, 2002; Tar-asov et al., 2018)…” Incorrect reference format.

Author Response

(The authors gave the same response as above.)
